# Policy Transfer via Latent Graph Planning

## Abstract

We introduce a transfer learning framework for deep reinforcement learning that integrates graph-based planning with self-supervised representation learning to efficiently transfer knowledge across tasks. While standard reinforcement learning aims to learn policies capable of solving long-horizon tasks, the resulting policies often fail to generalize to novel tasks and environments. Our approach addresses this limitation by decomposing long-horizon tasks into sequences of transferable short-horizon tasks modeled by goal-conditioned policies. We utilize a planning graph to generate fine-grained sub-goals that guide these short-horizon policies to solve novel long-horizon tasks. Experimental results show that our method improves sample efficiency and demonstrates an improved ability to solve sparse-reward and long-horizon tasks compared to baseline methods in challenging single-agent and multi-agent scenarios. In particular, compared to the state-of-the-art, our method achieves the same or better expected policy reward while requiring fewer training samples when learning novel tasks.

## 1 Introduction

Reinforcement learning (RL) has demonstrated impressive success in various challenging domains, including robotics (Lu et al., 2021), game playing (Vinyals et al., 2017), healthcare (Abdellatif et al., 2021), and conversational agents (Ouyang et al., 2022). The ability of RL agents to autonomously learn policies through trial-and-error has made them well-suited for tasks where predefined strategies are difficult to design. However, despite these advancements, RL often struggles when applied to long-horizon tasks where agents must learn a complex, extended sequence of behaviors. Two major challenges arise in these scenarios: effective exploration, which is difficult over long horizons as the number of possible state-action sequences grows exponentially; and credit assignment, where it is unclear which actions contributed to task success or failure (Arumugam et al., 2021).

Transfer learning can be used to mitigate these challenges by leveraging knowledge gained from a source task to accelerate learning in a related target task (Zhu et al., 2023). Similarities in structures or features between related tasks allow learned insights to be transferred, avoiding the need to start learning from scratch in each new task. However, these methods are often less successful as the task horizon increases, owing to the combinatorial explosion of potential action sequences and the compounding of errors over time (Gupta et al., 2019; Jiang et al., 2024).

In this paper, we propose a novel solution to this problem by automatically decomposing long-horizon tasks into sequences of short-horizon tasks, which are solved using a goal-conditioned policy. By focusing on short-horizon tasks, we reduce the complexity of the task space, making it easier for the policy to adapt to new but related tasks. We decompose each task by learning a latent space using self-supervised temporal contrastive learning, where states that are temporally and spatially close are mapped to nearby points in the latent space. We cluster the latent space to construct a graph that captures the relationship between different states. This latent space graph is used to plan a sequence of sub-goals to reach any desired temporally extended goal, and is used to guide the short-horizon goal-conditioned policy (see Fig. 1). While task decomposition has been extensively studied in prior works (Nasiriany et al., 2019; Huang et al., 2019; Hoang et al., 2021) to enhance performance within a single task, we show that such decompositions also significantly improve a policy's generalizability to novel tasks and lead to state-of-the-art transfer learning performance.

Our contributions are as follows: 1) We introduce a method for learning a latent space graph which can be used to automatically decompose a task into a sequence of shorter sub-tasks via planning.

Figure 1: Our approach involves training in a source environment by completing randomly sampled short-term goals (step A). We then iteratively roll out the partially trained policy to learn a latent space that captures the temporal structure of trajectories and the long-term task graph from a single expert demonstration (step B). To apply the policy to a new task, we fine-tune the short-term policy. This allows for effective transfer as we only need to fine-tune short-term goals, while the new long-term task is represented in the task graph from a single expert demonstration.

2) We empirically show in both single-agent and multi-agent reinforcement learning tasks that our approach learns generalizable policies that can be readily adapted to novel tasks, significantly improving policy convergence speed when compared to state-of-the-art transfer learning methods. 3) In the special case of transferring policies between isomorphic tasks, our approach allows for zero-shot transfer, only requiring edits to the planning graph while being able to re-use the underlying policy directly.

## 2 RELATED WORKS

### 2.1 GOAL-CONDITIONED REINFORCEMENT LEARNING

Goal-conditioned reinforcement learning (GCRL) is a framework where an agent learns to achieve a specified goal state instead of maximizing a scalar reward signal. Schaul et al. (2015) introduced the concept of universal value function approximators (UVFA), which extends the standard value function to consider goal states. Andrychowicz et al. (2017) proposed Hindsight Experience Replay (HER), a technique that allows the agent to learn from failures by treating the achieved state as the desired goal state. Recent works have extended GCRL to handle multi-goal scenarios (Plappert et al., 2018) and hierarchical goal-setting (Nachum et al., 2018; Levy et al., 2017).

Exploration is crucial for GCRL, especially in sparse reward settings. Go-Explore (Ecoffet et al., 2019) addresses this by building an archive of diverse, high-performing states during exploration and learning a policy to reach these states reliably. Skew-Fit (Pong et al., 2019) introduces a goal sampling scheme that favors goals of intermediate difficulty, encouraging exploration and learning. DISCERN (Li et al., 2021) learns a goal-conditioned policy using an unsupervised reward function that promotes exploration and skill discovery. Plan2Explore (Sekar et al., 2020), LEXA (Mendonca et al., 2021) and PEG (Hu et al., 2023) build on top DreamerV2 (Hafner et al., 2020) and promote exploration during training.

### 2.2 CONTRASTIVE REPRESENTATION LEARNING IN ROBOTICS

Contrastive learning has been successfully applied to robotics for learning state and reward representations. Laskin et al. (2020a) proposed the Contrastive Unsupervised Representations for Reinforcement Learning (CURL) framework, which learns a contrastive representation of raw pixels to improve sample efficiency in robotic control tasks. Zhan et al. (2022) introduced a framework for learning robotic manipulation skills using contrastive learning, demonstrating improved performance and generalization. Other works have utilized contrastive learning for various aspects of robotic learning. Singh et al. (2020) employed contrastive learning to learn reward functions, while Laskin et al. (2020b) used it to learn invariant representations. Florence et al. (2018) and Cao et al. (2022) trained view-angle invariant contrastive representations to improve robotic manipulation tasks, enabling the agent to handle variations in object poses and camera viewpoints. Cao et al. (2023) proposed a method for learning sim-to-real pixel-to-pixel consistent contrastive repre-

sentations, which allows for zero-shot transfer of policies learned in simulation to real-world robotic manipulation tasks. Park et al. (2024) and Park et al. (2024) used contrastive learning to learn a mapping from states to latent representation that preserves the temporal structure.

### 2.3 HIERARCHICAL REINFORCEMENT LEARNING

Hierarchical reinforcement learning (HRL) aims to learn a hierarchy of policies operating at different abstraction levels. The goal is to break down a complex task into simpler subtasks, which can be learned more efficiently. Sutton et al. Sutton et al. (1999) introduced the options framework, which extends the standard MDP to include temporally extended actions. Bacon et al. Bacon et al. (2017) proposed the Option-Critic architecture, which simultaneously learns the policy over options and the options themselves. Recent works have explored learning goal-conditioned hierarchical policies (Nachum et al., 2018; Levy et al., 2017) and combining HRL with meta-learning (Frans et al., 2017).

### 2.4 TRANSFER LEARNING IN REINFORCEMENT LEARNING

Transfer learning in RL aims to leverage knowledge learned from one task to improve learning efficiency and performance in another related task. Zhu et al. (2023) provides a comprehensive survey of transfer learning methods in RL. Rusu et al. (2016) introduced the Progressive Neural Networks (PNN) architecture, which allows for transferring knowledge across a sequence of tasks while avoiding catastrophic forgetting. Other approaches include learning invariant feature spaces (Gupta et al., 2017), meta-learning for fast adaptation (Finn et al., 2017), and learning transferable representations (Higgins et al., 2017).

Recent advancements include Distilling Policy Distillation (Czarnecki et al., 2019), which combines policy distillation with teacher-student curriculum learning for efficient knowledge transfer, and Kickstarting Deep Reinforcement Learning (Schmitt et al., 2018), which uses human demonstrations in a source task to initialize policies in a target task, reducing exploration and improving learning efficiency. JumpstartRL (Uchendu et al., 2023) uses a guidance policy to help a new policy to learn in a curriculum setting.

## 3 METHOD

In this section, we introduce our approach for transferring a policy from a source to a target task in a sample-efficient manner (also see Fig. 1). Section 3.1 introduces the training of our initial goal-conditioned policy (GCRL) executing randomly sampled short-horizon tasks in the source domain. The key to our method is that utilizing a policy that executes simple, short-horizon tasks will be easier to transfer than a policy handling long-horizon tasks directly. In section 3.2, we highlight how a sequence of sub-goals for a particular long-horizon task is created, namely our planning graph, given only a single expert demonstration of the desired task. This graph operates over a learned latent space covering the agent's behavior using contrastive learning to capture the temporal structure of the agent's trajectories. Finally, in section 3.3, we highlight how the sub-goals for the novel task are selected.

### 3.1 GOAL-CONDITIONED REINFORCEMENT LEARNING AGENT

Given an expert trajectory $\tau_{\text{expert}}$ for the long-horizon task in the source environment, we train a short-horizon goal-conditioned policy capable of completing navigation tasks with goals in close proximity to the starting point. To ensure that the GCRL agent adheres to the demonstrated task, we sample random starting states $s_0$ from the target trajectory and extract feasible short-term goals by short random walks. In particular, we sample the initial state from the expert trajectory $\tau_{\text{expert}}$ and sample a goal state $g \sim P(g|s_0)$. For a comprehensive algorithm description, we refer the reader to (Schaul et al., 2015) and (Schulman et al., 2017). We train our GCRL agent with the universal value approximator (Schaul et al., 2015) and Proximal Policy Optimization (Schulman et al., 2017).

## 3.2 TEMPORAL CONTRASTIVE LEARNING AND CLUSTERING

Providing sub-goals guiding the GCRL agents to complete tasks in target environments allows for efficiently transferring skills learned in the source environment to the target environment. However, this depends on the ability to provide accurate sub-goals to the GCRL agent. To achieve this, we utilize contrastive learning to distill a latent space representing temporal distances, specifically, the minimal steps required for an agent to transition from one state to another. However, obtaining the minimal temporal distance between state pairs requires optimal control between every pair of states. Hence, we use state pairs and corresponding temporal distances from rollouts generated by the GCRL agent for approximation. As these temporal distances may still being noisy, we employ the InfoNCE (Oord et al., 2018) approach to learn a mapping $f_w$ from the observational space to the embedding space, where geometric proximities in the embedding space mirror temporal distances in the trajectories. This relationship is encapsulated in Equation 1, with $d(\cdot, \cdot)$ representing a metric distance function. We choose $d(\cdot, \cdot)$ as the L2 distance. Adopting a metric space as $d(\cdot, \cdot)$ enables estimating temporal distances between unobserved state pairs using the triangular inequality. This contrastive learning and metric formulation, coupled with neural network modeling, allows our system to process and generalize from noisy trajectory data. During training, we select state pairs within $T$ timesteps in a trajectory to be positive samples and randomly sample states within the same batch to be negative samples. $T$ is a hyper-parameter governing the maximum temporal threshold for positive sample pairs.

$$L_{\text{tc}}(x, x_{pos}, X) = -\mathbb{E}\left[\log \frac{exp(-d(f_w(x), f_w(x_{pos})))}{\sum_{x' \in X} exp(-d(f_w(x), f_w(x')))}\right] \tag{1}$$

Note that the learned latent space reflects the temporal distances of the underlying trajectories used for training. Thus, curating a dataset representative of the state and transition distribution for the designated task is crucial. Collecting rollouts of states relevant to the desired task with temporal distances close to the minimal temporal distances is essential for learning latent space structures useful for the task.

In Algorithm 1, we sample initial states from an expert trajectory $\tau_{\text{expert}}$ to ensure we efficiently cover state regions relevant to the completing the task; we use the trained GCRL agent $\pi_\theta$ to collect rollouts; furthermore, we sample state pairs to balance the probabilities of sampling each state. After learning the temporal embeddings, we construct a graph to capture the essential temporal structure of the task. The graph is constructed as follows: first, we employ the K-means clustering algorithm to group the embeddings into distinct clusters and utilize the elbow method to determine the optimal number of clusters (Lloyd, 1982; Bengfort & Bilbro, 2019). Each cluster in the embedding space represents a node in the graph. Then, we create edges between nodes based on the observed transitions between clusters in the expert trajectory. Specifically, for each consecutive pair of states in the expert trajectory, we identify their corresponding clusters and add an edge between the associated nodes in the graph. It is crucial to note that the learned embeddings and the resulting graph are grounded in the original state space, enabling us to map each state to its corresponding embedding, cluster, and graph node. This property allows for seamless integration of the graph-based planning with the reinforcement learning agent. The constructed graph captures the essential temporal structure of the task, facilitating efficient planning and sub-goal generation for the agent during the transfer learning process.

## 3.3 TASK EXECUTION

After finetuning on the target environment, we combine the GCRL agent $\pi_\tau$, the temporal contrastive mapping $f_w$, the expert demonstration $\tau_{\text{expert}}$, and the cluster classifier to execute tasks. As shown in Algorithm 2, on each step, we predict the current cluster and select the next sub-goals $g$ as the state that transitions to the next cluster on the shortest path from the current cluster to the target cluster, or the target state if we are already in the target cluster, and execute the action sampled from $\pi_\theta(s, g)$.

---

**Algorithm 1** Training Temporal Latent Space

---

1: **Input:** env, $f_w$, $\pi_\theta$, $\tau_{\text{expert}}$, $P(g|s_0)$
2: $s_0 \sim \tau_{\text{expert}}$
3: $g \sim P(g|s_0)$
4: Dataset $\leftarrow$ rollouts($\pi_\theta$, env, $s_0$, $g$)
5: **while** not converged **do**
6: $\quad$ $x, x_{pos}, X \leftarrow$ BalancedSampling(Dataset)
7: $\quad$ Optimize $L_{\text{tc}}(x, x_{pos}, X)$
8: **end while**
9: ClusterClassifier $\leftarrow$ Cluster $f_w$(Dataset)
10: PlanningGraph $\leftarrow$ construct_graph(Dataset, $f_w$, $\tau_{\text{expert}}$)

---

**Algorithm 2** Task Execution

---

1: **Input:** env, $\pi_\theta$, $\tau_{\text{expert}}$, $f_w$, ClusterClassifier
2: $s \leftarrow$ env.reset()
3: **while** not done **do**
4: $\quad$ $c \leftarrow$ ClusterClassifier($f_w(s)$)
5: $\quad$ $g \leftarrow$ GetSubGoal($f_w$, $c$, $\tau_{\text{expert}}$)
6: $\quad$ action $\sim \pi_\theta(s, g)$
7: $\quad$ $s$, done $\leftarrow$ env.step(action)
8: **end while**

---

## 4 EXPERIMENTS

The primary goal of these experiments is to address the following questions: 1) Can our method reduce sample complexity for transfer learning in both single-agent and multi-agent environments? 2) How does our approach compare to baseline models in terms of performance across different target environments? 3) Are the sub-goals generated by our method semantically meaningful? 4) Can we zero-shot transfer to isomorphic tasks by only adapting the task graph?

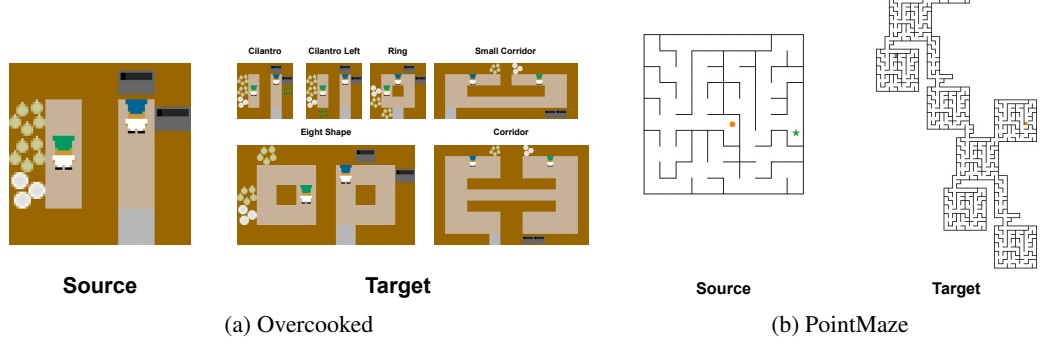

Figure 2: a) The source and target Overcooked (Carroll et al., 2019) tasks. The two chefs need to coordinate to make soup and deliver soups. In each environment, there are two chefs (the chef with the green hat and the chef with the blue hat), onion dispensers, plate dispensers, ovens (the grey box with a black top), a serving area (the plain light grey box), walls (brown box) and optionally cilantro dispensers. b) The source and target PointMaze (Pitis et al., 2020) tasks. Agents must navigate from the initial position (the orange point) to the target position (the green star).

## 4.1 SETUP

We evaluated our method against five baseline approaches across seven transfer learning tasks in the Overcooked environment (Carroll et al., 2019), a multi-agent, cooperative domain based on the video game *Overcooked*. Here, chefs must coordinate to prepare and deliver soups across varying kitchen layouts and recipe configurations. In this work, we focus on two-player scenarios where agents must coordinate to complete the high-level steps involved in preparing and serving soups, as outlined in Figure 3. To assess transfer learning performance, we pre-trained agents in a source environment, $env_s$, and subsequently transferred them to a set of target environments, $env_t$. The target environments were designed as variations of the source environment, differing either in layout or task complexity. For instance, the *Cilantro* and *Cilantro Left* environments introduce both new recipes and modified layouts, whereas environments such as *Ring*, *Eight Shape*, *Small Corridor*, and *Corridor* focus on increasingly complex layout configurations. These source and target environments are shown in Figure 2a. All experiments were conducted using partially observable agents (seeing the 3x3 grid centered at the agent). Each episode consisted of 500 timesteps, and agent performance was evaluated by the number of soups delivered per episode. The original Overcooked environment operates under deterministic dynamics with a fixed initial configuration. To introduce variability and prevent overfitting to the initial state, we randomized the agent's initial ten timesteps before policy execution. For our method, we provided a single expert trajectory for each target environment, generated through hand-crafted policies. We included baseline methods that have access to policies trained directly on the target environments for a fair comparison.

We compare our method against the following five baseline approaches:

- **No Transfer**: This approach trains an RL agent from scratch in the target environment, without utilizing any knowledge from the source environment.

- **Fine-tuning**: In this approach, an agent pre-trained in the source environment is fine-tuned in the target environment, allowing the agent to adapt its learned policies to the new task.

- **Policy Distillation (Loss)**: This method employs an auxiliary cross-entropy loss to align the action probabilities of a pre-trained policy from the source environment with the learning policy in the target environment (Schmitt et al., 2018).

- **Policy Distillation (Reward)**: This method uses a reward shaping term to incorporate the difference between the pre-trained critic from the source environment and the current policy's predictions at each timestep in the target environment (Czarnecki et al., 2019).

- **JumpStart RL (JSRL)**: This method begins by rolling out a guiding policy to assist the RL agent in moving closer to the goal (Uchendu et al., 2023). The number of steps the guiding policy is used depends on a curriculum schedule (e.g. gradually decreasing from 55 to 0). As training progresses, the RL agent gradually relies less on the guiding policy, allowing it to learn more independently. Several JSRL configurations were evaluated based on the following factors:

    - **Guiding Policy Source**:
        * *Source Environment*: The guiding policy is trained on the source environment.
        * *Target Environment (Oracle)*: The guiding policy is trained on the target environment, giving the agent an oracle-like advantage.
    - **Fine-tuning**:
        * *JSRL Tune*: The policy network is initialized from the source environment policy.
        * *No Fine-tuning*: The policy network is randomly initialized.

## 4.2 TRANSFER LEARNING RESULTS

We present the average number of soups delivered throughout training for each method in Figure 4, and report the convergence speeds and final performances in Table 1 and Table 2. Our method demonstrates a significant advantage in convergence speed, as shown in Table 1. Notably, our methods performs comparably or better when comparing to JSRL with access to the oracle policy, trained in the target environment, as the guiding policies.

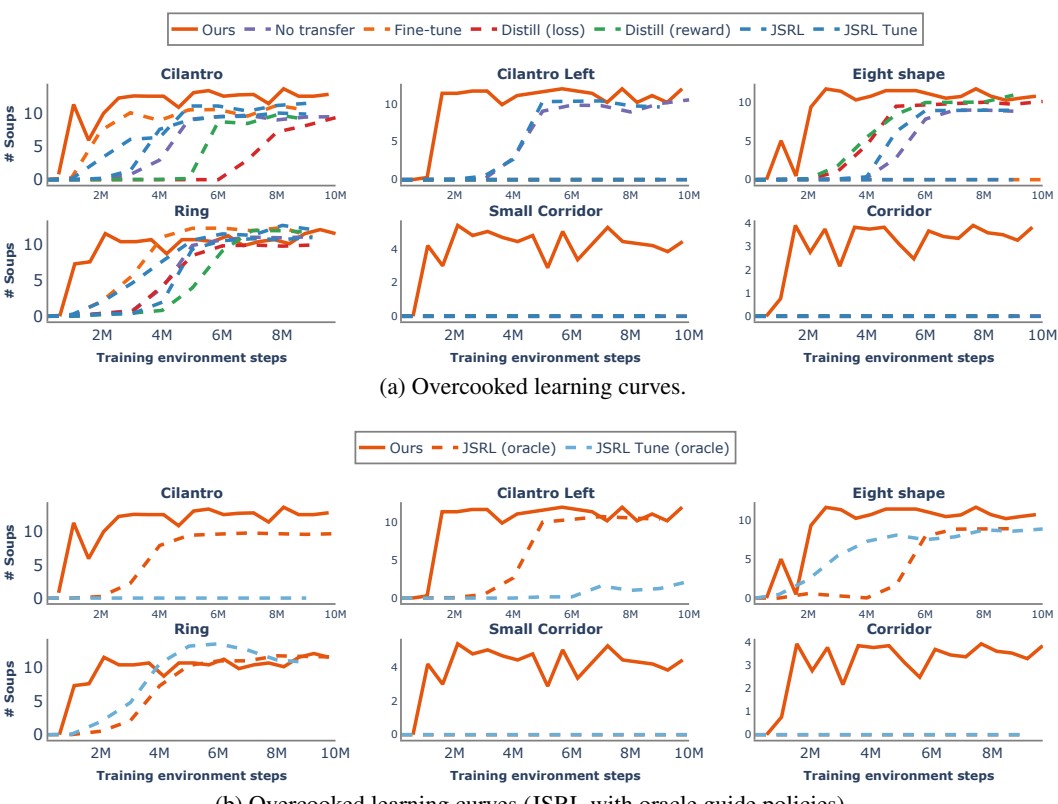

Figure 3: Overcooked recipes. To make one soup, the two chefs need to 1) fetch three onions from the onion dispenser and put them into the oven one by one, and 2) turn on the oven and wait for 20 steps, and 3) fetch a plate from the plate dispenser, take the soup from the oven to the plate, and 4) Optionally, to make a cilantro soup, fetch Cilantro from the dispenser and put it on the soup plate.

(a) Overcooked learning curves.

(b) Overcooked learning curves (JSRL with oracle guide policies).

Figure 4: Overcooked Learning Curves. Average soups delivered over 50 episodes throughout training. Note, baselines in *small corridor* and *corridor* do not deliver any soups, thus overlapping flat lines. a) compares our method with baselines; b) compares our method with JSRL where guiding policies are trained in the target environments.

In environments where the layouts remain similar but the recipes differ—such as *Cilantro* and *Cilantro Left*—our method consistently outperforms the baselines. Notably, transfer learning methods struggle in these settings and sometimes even perform worse than No Transfer. This is likely due to the inherent bias from the source environment's policies, which can hinder learning the subtle task differences in the target environment. For example, in environments with cilantro recipes, agents tend to follow the original recipe and fail to add cilantro to the soup before serving, leading to severe performance degradation. In contrast, our method effectively transfers to these environments, handling task-specific nuances that significantly impact performance.

In environments where the recipes remain similar but the layouts change—such as *Ring*, *Eight Shape*, *Small Corridor*, and *Corridor*—our method performs comparably or better than the baselines in most cases, while requiring fewer training samples. Interestingly, while JSRL with an oracle guiding policy baselines have an inherent advantage in these settings, our method still achieves superior or comparable results. This is especially evident in challenging environments like *Small Corridor* and *Corridor*, where other methods struggle to deliver any soups. The difficulty in these environments arises from the need for agent coordinations to avoid blocking each other in the narrow corridors. Our method excels in such scenarios, demonstrating its strength in transferring to long-horizon multi-agent planning and coordination tasks.

Overall, while baselines such as JSRL with an oracle guiding policy have inherent advantages, particularly in terms of access to more complete information, our method consistently outperforms them by better adapting to the intricacies of new environments with minimal additional training data.

| Environment | Cilantro | Cilantro Left | Eight Shape | Ring | Small Corridor | Corridor |
|:---:|:---:|:---:|:---:|:---:|:---:|:---:|
| **Ours** | 3.1M | 1.6M | **2.6M** | **2.1M** | **2.1M** | **1.6M** |
| No transfer | 5.0M | 6.0M | 7.0M | 6.0M | n/a | n/a |
| Fine-tune | **3.0M** | **1.0M** | n/a | 5.0M | n/a | n/a |
| Distill (loss) | 10.0M | 2.0M | 5.0M | 6.0M | n/a | n/a |
| Distill (reward) | 8.0M | 4.0M | 6.0M | 7.0M | n/a | n/a |
| JSRL | 5.3M | 6.0M | 6.0M | 5.8M | n/a | n/a |

Table 1: Overcooked training steps to convergence (reaching 90% of the max soups per method per environment) table. n/a means the method did not deliver any soup.

| Environment | Cilantro | Cilantro Left | Eight Shape | Ring | Small Corridor | Corridor |
|:---:|:---:|:---:|:---:|:---:|:---:|:---:|
| **Ours** | **13.72** | **12.00** | **11.76** | 12.04 | **5.40** | **3.92** |
| No transfer | 9.72 | 10.54 | 9.00 | 11.06 | 0.00 | 0.00 |
| Fine-tune | 11.22 | 0.02 | 0.00 | **12.32** | 0.00 | 0.00 |
| Distill (loss) | 9.36 | 0.02 | 10.12 | 9.90 | 0.00 | 0.00 |
| Distill (reward) | 9.80 | 0.04 | 10.94 | 11.92 | 0.00 | 0.00 |
| JSRL | 7.85 | 5.85 | 6.73 | 12.18 | 0.00 | 0.00 |

Table 2: Overcooked max soups delivered table.

### 4.3 SEMANTICALLY MEANINGFUL SUB-GOALS

The sub-goals generated from subsection 3.2 demonstrate a semantically meaningful breakdown of tasks, such as fetching onions, loading them into the oven, and serving soups, as shown qualitatively in Figure 5. This empirically shows that self-supervised temporal contrastive learning can discover meaningful task structures from rollouts. A possible explanation for this lies in the latent space clusters, which tend to form around bottleneck structures. These bottleneck transitions represent sequences of actions that allow the agent to reach previously inaccessible states, often corresponding to natural sub-goals. For instance, fetching an onion when the agent has none allows it to transition to states where it can carry onions, a task-critical sub-goal.

### 4.4 ZERO-SHOT TRANSFER BY ADAPTING TASK GRAPH

Our method enables efficient transfer to new environments when an isomorphic mapping exists between the source and target environments, allowing their structure to be adapted to fit the task graph representation. We specifically designed a transfer learning task in the Point Maze environment (Pitis et al., 2020) to exploit this capability. As shown in Figure 2b, Point Maze is a continuous 2D environment where the agent navigates from a randomly initialized position to a goal. The agent's observations consist of 2D lidar distance measurements and the displacement to the goal, while its actions are 2D planar velocities. The objective is to reach the goal.

To create the target environment, we expanded the source maze by copying and pasting sub-parts, constructing a larger maze. Since an isomorphic mapping between the source and target environ-

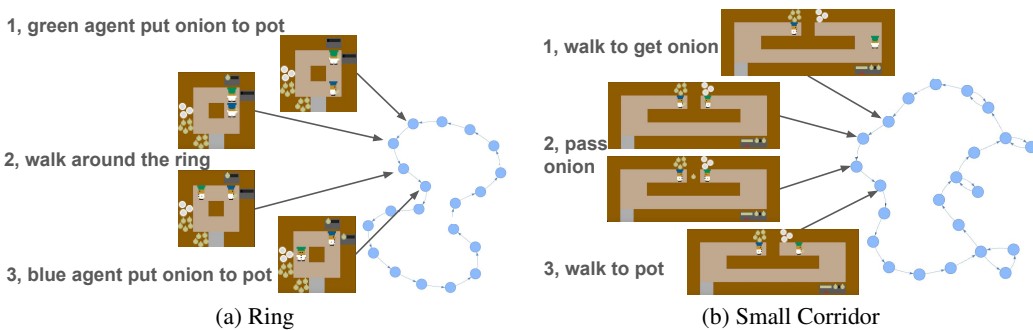

(a) Ring                                    (b) Small Corridor

Figure 5: Overcooked task graph and sample sub-goals. Learnt task graph and sample node transitions for overcooked environments. Semantically meaningful breakdown of the task emerges naturally from the temporal contrastive embedding clusters. For example, the sub-goals qualitatively demonstrate the intentions for handing over onions, fetching plates, putting onions into the oven, and taking soups out of the oven.

ments allowed our method to directly adapt the learned task graph, transfer to the target environment was achieved without additional learning. This structural similarity enabled us to bypass the training phase, leveraging the task graph to guide the agent's behavior in the new environment.

We evaluated each approach over 500 episodes and recorded the success rate. As shown in Figure 6, our method significantly outperforms baselines, achieving superior performance without requiring any training in the target environment.

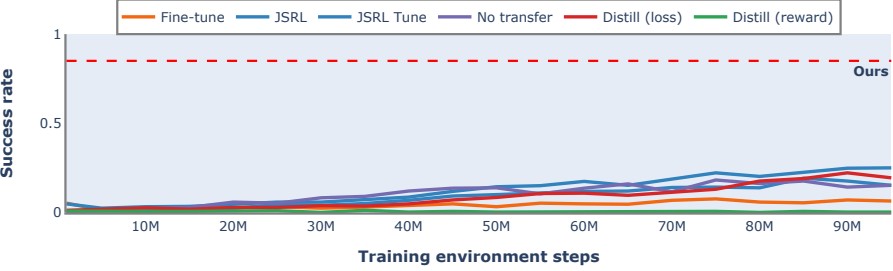

Figure 6: Point maze Transfer Learning Curve. The average success rate of reaching the goal is calculated over 500 episodes. Note that our method does not require training for this experiment.

## 5 CONCLUSION

This paper introduced a novel transfer learning framework for deep reinforcement learning that combines goal-conditioned policies with self-supervised learning of temporal abstractions. Experiments on Overcooked multi-agent coordination tasks demonstrated the effectiveness of our framework in terms of improved sample efficiency, the ability to solve sparse-reward and long-horizon challenges, and enhanced interpretability through the automatic discovery of meaningful sub-goals. These findings highlight the advantages of integrating goal-conditioned RL with self-supervised temporal abstraction learning for successful transfer to complex target domains, demonstrating superior performance compared to baseline methods such as fine-tuning, policy distillations, and curriculum learning methods. Compared to state-of-the-art baselines, our method achieves the same or better performances while requiring fewer training samples. Our work opens up exciting directions for future research, such as integrating language guidance into the contrastive learning process and applying our framework to real-world robotics tasks, paving the way for more intelligent, adaptable, and collaborative AI systems.

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
