# OpenReview forum: "Policy Transfer via Latent Graph Planning"
_ICLR.cc/2025/Conference — Submitted to ICLR 2025_

### Official Review · Reviewer_M9xG · 2024-10-30

**Soundness:** 2
**Presentation:** 2
**Contribution:** 2
**Rating:** 3
**Confidence:** 5

**Summary:**

This paper presents a novel framework for transfer learning in deep RL that combines goal-conditioned policies with self-supervised representation learning. The framework aims to improve sample efficiency and generalization to novel tasks, especially long-horizon tasks. The authors propose to decompose long-horizon tasks into sequences of short-horizon tasks represented by a graph. This graph, constructed using self-supervised temporal contrastive learning and clustering, guides a goal-conditioned policy to efficiently solve new tasks. The method is evaluated in single-agent and multi-agent environments, demonstrating superior performance compared to baseline approaches like fine-tuning, policy distillation, and curriculum learning methods.

**Strengths:**

The paper presents a novel combination of goal-conditioned RL with self-supervised temporal contrastive learning for efficient transfer learning. This approach, particularly the use of a latent space graph for planning and sub-goal generation, shows originality in addressing the challenges of transfer learning in long-horizon tasks. The paper is generally well-written and easy to follow. The authors provide a clear explanation of the proposed framework, including the training process, task execution, and the rationale behind their design choices. The use of figures effectively illustrates the key concepts and the overall workflow.

**Weaknesses:**

While the experimental results show promise, there are concerns regarding the evaluation methodology and the strength of the evidence supporting the central claims. For instance, the evaluation primarily focuses on the Overcooked environment, and the comparison with baseline methods, particularly JSRL, raises questions about the fairness of the evaluation. The authors claim their method performs comparably or better than JSRL, even when JSRL has access to an oracle policy trained on the target environment. This result seems surprising and requires further investigation and justification. It is especially surprising that oracle is underperforming by a big margin (Figure 4).

While the idea of decomposing long-horizon tasks into shorter sub-tasks is not entirely new, the paper's application of this concept to transfer learning and the use of self-supervised temporal contrastive learning for graph construction contribute to the novelty of the approach. However, the evaluation primarily focuses on relatively simple tasks in the Overcooked environment, and it is unclear how well the method would generalize to more complex and diverse domains. To strengthen the paper's impact, the authors should consider evaluating their framework on a wider range of tasks and environments, demonstrating its broader applicability and robustness.

The paper lacks a comprehensive ablation study to analyze the contributions of individual components within the framework. For example, evaluating the performance of the goal-conditioned policy without the latent graph planning, or assessing the impact of different contrastive learning methods or clustering algorithms on the quality of the generated sub-goals, would provide valuable insights into the effectiveness and robustness of the proposed approach.

The Experiments Section lacks crucial details for a comprehensive evaluation and reproducibility. For example:
- The paper does not mention the number of seeds used during training. Reporting results from multiple random seeds is essential for assessing the statistical significance of the findings and ensuring that the observed performance is not due to chance or a particular random initialization.
- The paper does not report confidence intervals for the results. Confidence intervals provide a measure of uncertainty around the reported averages, allowing readers to assess the reliability of the results and the potential variability across different runs.
- The paper does not provide explanations for hyperparameters or network architectures used to obtain the results. Also, there are no mentions of releasing the source code.

The absence of this information raises **serious** concerns about the robustness and reproducibility of the results, making it difficult to assess the statistical significance and the generalizability of the findings.

Also, there are no discussions of the possible limitations of the work. It could include discussions about how and where the proposed method fails or underperforms. Expanding on this with concrete examples and potential mitigation strategies would strengthen the paper.

Last but not least, there are formatting issues in the paper. For instance, Table captions should be on top while they are on the bottom for Tables 1 and 2.

**Questions:**

How would the proposed method perform in environments with continuous control, more complex agent dynamics, or significantly different task structures compared to Overcooked? Providing evidence of the framework's ability to generalize beyond the specific environment used in the evaluation would strengthen the paper's claims.

As already mentioned in the Weaknesses, comparison with JSRL needs further investigation. Can the authors elaborate on the performance comparison with JSRL, particularly when JSRL uses an oracle policy? A more detailed analysis, considering factors like the quality of sub-goals and the impact of oracle guidance, would clarify the relative strengths and weaknesses of the two approaches.

Can the authors provide an ablation study to isolate the contributions of individual components within the framework? This analysis would help to understand the importance of each component and the overall robustness of the proposed approach.

---

### Official Review · Reviewer_rz6Z · 2024-10-31

**Soundness:** 3
**Presentation:** 2
**Contribution:** 2
**Rating:** 5
**Confidence:** 4

**Summary:**

This paper proposes a framework to transfer knowledge acquired from one task to another through sampling in a latent graph constructed by clustering states/subgoals in the embedding space. The author(s) introduced their method by comprehensively explaining each component of their architecture and then showed its effectiveness with a good empirical study.

**Strengths:**

1. This paper is well-structured. The authors first analyze the common limitations of existing RL approaches when transferring knowledge and then describe how their proposed method solved the mentioned problem, making this paper understandable.

2. The results in the experiments part show that the architecture is promising. The selection of baselines and environments is reasonable and related to the theme of this paper.

**Weaknesses:**

1. A large proportion of the architecture is implemented by existing methods and the whole architecture seems like a combination of previous works. There are not enough original contributions.

2. All the experiments are done in the Overcooked environment. The author(s) didn't show the proposed method's performance in other widely used environments, such as MuJoCo or real-life simulation tasks. Also, some comparisons with SOTA methods, such as HRAC and HIGL, are expected.

3. To provide more statistical information, it is better to add error bars or confidence regions to the curves.

4. Detailed experimental settings such as hyperparameters are missing, reducing the reproducibility of the results.

5. Proving the effectiveness of the proposed architecture purely by experimental results is completely OK. However, the paper would be more convincing if the author(s) could provide some theoretical analysis of how and why the method is good/superior to previous works.

**Questions:**

1. In part 4.1, for no-transfer and fine-tuning, are the agents hierarchical? Do they consist of two or more levels of agents or just a non-hierarchical agent trained by traditional DRL methods such as PPO?

2. Following the previous question, what method is used to train the high/low-level agent if the agent is hierarchical?

3. In section 3.3, I would like to know how the function GetSubGoal works. Could the authors provide more details about how this function samples the next subgoal?

4. From my understanding of the paper, the method relies on expert demonstrations for initial exploration. However, expert data rarely exists in many environments. Could the author(s) demonstrate how will the method perform if no expert data is available?

5. (This may coincide with question 3) Could the author(s) elaborate a little bit more on how the graph-based planning is performed?

---

### Official Review · Reviewer_PVWH · 2024-11-03

**Soundness:** 3
**Presentation:** 3
**Contribution:** 2
**Rating:** 3
**Confidence:** 4

**Summary:**

The paper proposes to solve long-horizon, sparse reward problems by transferring knowledge from pre-trained goal-conditioned agents and combining them according to the directions of a planning graph. The proposed method is evaluated in both single and multi-agent scenarios demonstrating its effectiveness.

**Strengths:**

- The overall approach is well-motivated and clearly stated.
- Empirical evaluations look good.

**Weaknesses:**

1. Discussion on state clustering details: From Figure 1 and the description of the pipeline in Section 3.2, latent states are clustered into high-level nodes in the task graph. This is related to the state abstraction literature, which could have naturally led to more discussion on the clustering quality and the measures taken to ensure the clustering quality.
2. Missing curriculum learning literature and baselines: similar things happened to curriculum learning. The paper discusses resuing agents trained in source tasks to solve target tasks without a thorough treatment of curriculum learning literature. This also may lead to the lack of evaluations against popular curriculum learning baselines like Goal-GAN and Prioritized Level Replay.
3. Novelty concerns: The concept of using task graphs to guide agents solve long-horizon tasks is not new. As also mentioned by the authors, they only show that "such decompositions also significantly improve a policy's generalizability to novel tasks". Thus, I am not sure if it is proper to claim learning latent space graph which is then used to decompose a task as one of the contributions.

**Questions:**

1. In section 3.2, it's stated that "geometric proximities in the embedding space mirror temporal distances in the trajectories". I wonder if visual similarities can always translate to temporal proximities. For example, two states near a fire pit may look alike visually to the mapping $f_w$ even though it may take multiple time steps to circumnavigate the fire pit. If not, I would like to see some guarantees or empirical verifications on whether $f_w$ can faithfully capture the temporal distance in more complex environments like this.
2. What are those small loops in Figure 5 (b)? And how is it possible that the overall graph is a big loop, i.e., does it indicate that all action consequences are reversible? I suppose it's not since you cannot cook the onion soup, serve it, and then "uncook" it back to raw onion...

---

### Official Review · Reviewer_Vf7L · 2024-11-04

**Soundness:** 2
**Presentation:** 2
**Contribution:** 2
**Rating:** 3
**Confidence:** 3

**Summary:**

The paper introduces a novel framework for policy transfer in deep reinforcement learning that capitalizes on graph-based planning and self-supervised representation learning to facilitate knowledge transfer across tasks. Specifically, the framework begins by training a Goal-Conditioned Reinforcement Learning (GCRL) agent using a single expert demonstration. Additional data collected with this pre-trained agent is then used to learn a latent representation. Clusters identified via KMeans serve as nodes of the latent graph, with edges representing transitions between states from the expert trajectory that map to these clusters. In downstream tasks, sub-goals derived from this graph guide the pre-trained GCRL agent, improving its ability to address target tasks efficiently. The authors evaluate their method in both single-agent and multi-agent scenarios, demonstrating enhanced sample efficiency and superior performance in solving sparse-reward tasks compared to conventional baseline methods.

**Strengths:**

1. The paper provides extensive empirical evidence demonstrating the effectiveness of the proposed method over considered baseline methods.

**Weaknesses:**

- Clarity and Detail in Methodology:
  1. Is the GCRL agent trained with only one expert trajectory in Section 3.1? How many transitions are there?
  2. Where is the utilization and integration of temporal distances within the latent graph construction?
  3. What does this sentence mean and how is this guaranteed in this work, "Collecting rollouts of states relevant to the desired task with temporal distances close to the minimal temporal distances is essential for learning latent space structures useful for the task"?
  4. This sentence is not clear to me "we sample state pairs to balance the probabilities of sampling each state", could the authors explain what did you do here?
  5. Is the latent graph build soly based on the one expert trajectory used in section 3.1?
- Comparison with Hierarchical Methods:
  1. The comparison of the proposed hierarchical, graph-based method with primarily flat learning methods may not adequately represent its advantages or limitations. Could the authors provide comparison with hierarchical goal-conditioned transfer learning methods that do not build a graph explicitly?
- Handling of Task Stochasticity:
  1. It remains unclear whether the tasks considered are deterministic. The paper should address how the proposed method performs under stochastic conditions, which are common in real-world applications.

**Questions:**

1. How effectively does the KMeans algorithm perform when applied to the observation space compared with latent space on the considered tasks in the paper?
2. The latent graph appears to be undirected; how do the authors address the presence of cycles within the graph, if any exist?

---

### Meta-Review · Area_Chair_7YG2 · 2024-12-24

**Metareview:**

This paper presents a framework that uses goal-conditioned RL, self-supervised temporal contrastive learning, and clustering to build a latent graph for sub-goal generation. Experiments show speedups in learning transfer and improved long-horizon task performance.

The method has reasonable assumptions and motivation to decompose tasks via sub-goals. It is demonstrated to show improvements in sparse-reward, multi-agent settings, along with straightforward integration of planning and contrastive learning.

Experiments are shown in the Overcooked environment with limited comparisons in other environments. There are missing ablations and analyses on the clustering and contrastive learning components. Reviewers also raised concerns about reproducibility.

**Additional Comments On Reviewer Discussion:**

The idea of leveraging latent graphs for sub-goal planning is promising, but there are concerns about limited experimental breadth, missing baselines, and insufficient details, even after the rebuttal phase. I would encourage the authors to address these concerns and resubmit.

---

### Decision · Program_Chairs · 2025-01-22

Reject